# Robust Automated Mouse Micro-CT Segmentation Using Swin UNEt TRansformers

**DOI:** 10.3390/bioengineering11121255

**Published:** 2024-12-11

**Authors:** Lu Jiang, Di Xu, Qifan Xu, Arion Chatziioannou, Keisuke S. Iwamoto, Susanta Hui, Ke Sheng

**Affiliations:** 1Department of Radiation Oncology, University of California San Francisco, San Francisco, CA 94115, USA; lu.jiang@ucsf.edu (L.J.);; 2Department of Molecular and Medical Pharmacology, University of California Los Angeles, Los Angeles, CA 90095, USA; 3Department of Radiation Oncology, University of California Los Angeles, Los Angeles, CA 90095, USA; 4Department of Radiation Oncology, City of Hope, Duarte, CA 91010, USA

**Keywords:** micro-CT, mouse, organ segmentation, deep learning, Swin Transformers

## Abstract

Image-guided mouse irradiation is essential to understand interventions involving radiation prior to human studies. Our objective is to employ Swin UNEt TRansformers (Swin UNETR) to segment native micro-CT and contrast-enhanced micro-CT scans and benchmark the results against 3D no-new-Net (nnU-Net). Swin UNETR reformulates mouse organ segmentation as a sequence-to-sequence prediction task using a hierarchical Swin Transformer encoder to extract features at five resolution levels, and it connects to a Fully Convolutional Neural Network (FCNN)-based decoder via skip connections. The models were trained and evaluated on open datasets, with data separation based on individual mice. Further evaluation on an external mouse dataset acquired on a different micro-CT with lower kVp and higher imaging noise was also employed to assess model robustness and generalizability. The results indicate that Swin UNETR consistently outperforms nnU-Net and AIMOS in terms of the average dice similarity coefficient (DSC) and the Hausdorff distance (HD95p), except in two mice for intestine contouring. This superior performance is especially evident in the external dataset, confirming the model’s robustness to variations in imaging conditions, including noise and quality, and thereby positioning Swin UNETR as a highly generalizable and efficient tool for automated contouring in pre-clinical workflows.

## 1. Introduction

Radiotherapy (RT) treats an estimated 52.4% of all cancer patients [1] and contributes to 40% of cures [2]. Despite the long history of RT, its clinical applications are still being rapidly developed with ever-improving technology, understanding of underlying biology, and emerging combined therapy [3,4,5,6]. The pre-clinical study of small animal RT response is a pivotal step to bridge the gap between in vitro concepts and clinical practice [7,8]. With the recent advances in gene knockout and transgenic techniques, mice and other rodents have become widely used model organisms of choice in pre-clinical research given their considerable similarity to human physiology and pathology, versatility in genetic modification, cost-efficiency, and availability [9,10].

The past decade has continued to witness the development of RT prompted by novel small animal irradiator-enabled comprehensive studies [11]. Small animal irradiators are commonly designed with kilovolt X-ray radiation sources combined with high-resolution 3D image guidance in the form of onboard micro-computed tomography (CT), the latter of which is used for treatment planning and target localization parallel to modern image-guided human radiotherapy workflow [11,12]. A major step in recent pre-clinical devices is the development of image-guided small animal irradiators, which introduces whole-body imaging modalities and advanced image-guided irradiation systems for small animals [13,14,15]. The Small Animal Radiation Research Platform (SARRP, Xstrahl Ltd., 480 Brogdon Road, Suite 300 Suwanee, GA 30024, USA) and X-Rad SmART (Precision X-Ray Inc., 14 New Road Madison, CT 06443, USA) are two commercial image-guided small animal radiation treatment systems [16,17]. Image-guided small animal irradiators empower more accurate radiation dose characterization and delivery. However, the workflow for image-guided small animal irradiation traditionally requires manual organ delineation. Although staffed by skilled researchers, the practicality of manual segmentation can be challenging due to limited time, resources, and the specific training required to perform manual delineation [18,19]. As a result, organ contouring is often abbreviated, inaccurately performed, or even bypassed. Subsequently, the 3D dose is inadequately characterized, compromising the experimental reproducibility and translatability. Draeger et al. highlighted this issue, pointing out the lack of detailed reporting on essential experimental physics and dosimetry across radiobiology studies, which critically affects the replicability and comparability of research findings in the field [20].

Apart from the technical difficulty, manual mouse organ segmentation faces additional challenges. First, the image-guided mouse irradiation workflow must be accomplished in real time with the animal under anesthesia, compared to asynchronous human treatment planning. Second, a small animal irradiation experiment is typically conducted by one or two researchers overseeing all aspects of the experiments, including animal preparation, anesthesia, treatment planning, dosimetry, and machine operations, whereas human therapy is performed by a team, with each operator focusing on a specific link of the process to limit error and variation. Third, micro-CT images for small animal irradiation are significantly noisier than high-resolution diagnostic CT for human patient treatment planning. Fourth, the quality of mouse micro-CT images can vary considerably due to differences in imaging methodologies and setups across various facilities. Additionally, delineating the tumor can be particularly challenging depending on its location, especially in orthotopic grafts, making the a priori knowledge of normal organ anatomy crucial. Failing to recognize and address these challenges prevents biologists from extracting the essential dose–volume information and hampers the development of more advanced pre-clinical irradiation techniques, such as intensity-modulated radiotherapy (IMRT).

Several pioneering methods have been developed to automate small animal organ contouring. For instance, atlas-based segmentation was used for mouse whole-body imaging [21,22,23,24]. However, its segmentation quality relies on deformable registration and the prior anatomical knowledge defined in the atlas. For abdominal organs, the performance is moderate (60–80% DSC), while for challenging organs like the spleen, which lacks clear boundaries with adjacent tissues, it is unacceptably low (below 50% DSC), often requiring extensive manual corrections. Following that, Van der Heyden et al. [25] developed a multi-atlas-based image segmentation (MABIS) algorithm for six organs to account for individual variations and enhance low-contrast organ segmentation. However, their proposed post-processing techniques are manual and time-demanding (~12 min per mouse). In addition to atlas-based approaches, Akselrod-Ballin et al. [26] proposed super-pixel machine learning algorithms learned from multiple imaging modality inputs, which can be generalized to various tissues and imaging modalities but added additional complexity to data acquisitions and required dedicated animal holders.

Deep learning (DL) has shown great promise in image processing in the past decade, including segmentation [27,28,29]. Multiple DL approaches were proposed for the task of mouse segmentation. Specifically, Van der Heyden et al. [30] designed a 3D U-Net to automatically contour mouse skeletal muscles. Wang et al. [31] developed a 3D two-stage deeply supervised convolutional neural network (CNN) to segment multiple major organs, but the spleen segmentation accuracy was low at 57% DSC. AIMOS [32] (AI-based Mouse Organ Segmentation) was designed to be fully automatic, with several 2D U-Net-like architectures that differ in the number of encoding and decoding levels, but the 2D architectures lack inter-slice continuity. Malimban et al. [33] applied several no-new-Net (nnU-Net) variants to segment mouse micro-CT scans and found that 3D nnU-Net models outperformed 2D models and AIMOS, but their study only focused on the clear-boundary thorax region. Lappas et al. [34] proposed a preprocessing step that converted Hounsfield units (HUs) to mass density to improve dataset consistency, followed by a 3D U-Net for mouse micro-CT segmentation, but no results were reported for challenging abdominal organs.

Fully convolutional neural networks (FCNNs), such as U-Net, have demonstrated solid performance in various medical image segmentation tasks [35,36,37]. However, these methods are not built on the inherent self-attention mechanism and are unstable when segmenting heterogeneous data outside of the training cohort. Studies [38,39] have shown that FCNNs often face challenges in certain tasks, such as brain tumor and breast tumor segmentation, achieving suboptimal volume-based and surface-based metrics, where data heterogeneity and domain shifts significantly affect performance. The challenge is more significant for pre-clinical images due to the lack of standardization. Recently, Transformers, leveraging parallel learning and attention mechanisms, have demonstrated efficient and robust inferences in computer vision tasks [39,40,41,42]. Rolfe et al. presented an open-source Mouse Embryo Multi-Organ Segmentation (MEMOS), using a fused architecture of U-Net and Transformers (UNETR) [42,43]. More recently, Transformers evolved to be Swin Transformers [44,45,46,47]. These models applied the shifted windows self-attention scheme, referencing neighboring tokens during model propagation to enhance their regional learning ability with surrounding information. This approach makes the overall network architecture more robust and generalizable. Inspired by Swin Transformers, Swin UNEt TRansformers [48] (Swin UNETR) were developed specifically for generalizable medical imaging segmentation. The model outperformed other state-of-the-art approaches in brain tumor segmentation tasks, including nnU-Net [36], SegResNet [49], and a Vision Transformer-based model, TransBTS [50] by achieving higher Dice scores. However, the efficacy of Swin Transformers for pre-clinical micro-CT segmentation has not been studied.

Here, we introduce Swin Transformers for automatic major mouse organ contouring. Our model is trained and validated on a publicly available micro-CT dataset and compared with state-of-the-art models, the 3D U-Net architecture, and AIMOS. An in-house dataset, acquired using a different micro-CT at lower kVp, was employed to assess the generalizability of our method.

## 2. Materials and Methods

### 2.1. Data

In this study, we used public and in-house mouse micro-CT datasets. The public dataset [51] consisted of two types of scans: native micro-CT (NACT) and contrast-enhanced micro-CT (CECT). Specifically, the NACT dataset included 140 whole-body scans from 20 mice obtained at seven different time points using a pre-clinical micro-CT scanner (Tomoscope Duo, Erlangen, Germany) with an energy level of 65 kVp. The CECT dataset contained 81 scans from 8 mice acquired at various time points with an InSyTe micro-CT scanner (BMIF TriFoil Imaging, Dijon Cedex, France) at 75 kVp. Both types of public scans used an isotropic resolution of 0.14 mm. Additionally, we used an in-house dataset (IHCT) consisting of 5 scans from 5 different mice. These images were captured using an X-RAD SmART scanner (Precision X-Ray Inc., 14 New Road Madison, CT 06443, USA) at 40 kVp and featured an isotropic resolution of 0.2 mm. The public dataset annotations were performed solely by Biologist A and were used as provided. For the in-house dataset, Biologist B independently conducted all manual segmentations. This distinction between annotation sources allows for the generalizability assessment of our models across different institutions. The specifics for each dataset are further detailed in Table 1.

For model training on the NACT dataset, data was divided into training and validation sets consisting of 14 mice (98 scans), with testing withheld for 6 mice (42 scans). Similarly, the CECT dataset was divided into training and validation with 6 mice (66 scans) and testing involving 2 mice (15 scans). This division ensured that subsets for training, validation, and testing were strictly separate at the individual animal level, providing an unbiased and thorough evaluation of all models. The study targeted seven critical organs: the heart, lungs, liver, intestine, spleen, kidneys, and bladder.

All data were homogenized to ensure consistency as follows. Scan voxel values were normalized to [0, 1] for three models in the study. Mouse immobilization devices were manually contoured and then scrubbed from the background to ensure clean background for model training. The lower-resolution private dataset was linearly resampled to be 0.14 mm isotropically, like the public dataset. The same data augmentation techniques were applied to all models, including rotation, scaling, noise addition, contrast adjustments, and mirroring.

### 2.2. Model

#### 2.2.1. Swin Transformers for Semantic Segmentation

Swin Transformer, a variant of the general Transformers model, employs an efficient shifted window partitioning scheme, making it suited for medical image analysis where multi-scale feature extraction is important. In this study, Swin UNEt TRansformers (Swin UNETR) was adapted from Hatamizadeh et al. [48] Swin UNETR reformulates the segmentation task as a sequence-to-sequence prediction problem, where multimodal input data are projected into 1D sequences of embeddings, utilizing a hierarchical Swin Transformer as the encoder. This encoder has a patch size of 2 × 2 × 2 with 7 channels, resulting in a 56-dimensional feature space. The encoder is characterized by 4 stages with 2 transformer blocks in each, making a total of 8 layers. Swin UNETR has a U-shaped network design in which the extracted feature representations of the encoder are used in the decoder via skip connections at each resolution. At each stage of the encoder and bottleneck, the output feature representations are adjusted in size and fed into a residual block. This block consists of two 3 × 3 × 3 convolutional layers, normalized by instance normalization layers. Following this, the resolution of the feature maps is doubled using a deconvolutional layer, and the resultant outputs are concatenated with those from the preceding stage. The final segmentation is achieved using a 1 × 1 × 1 convolutional layer with a sigmoid activation function. The soft Dice loss function is applied in a voxel-wise manner. The Swin UNETR models were trained on NACT and CECT datasets separately. The inference window size is 128 × 128 × 128 with an overlap factor of 0.8 between windows. The U-shaped design incorporates Swin Transformer’s strengths into a structure conducive to complex segmentation tasks, such as mouse organ segmentation for micro-CT scans. More Swin UNETR architecture details can be found in Figure 1.

#### 2.2.2. nnU-Net

The nnU-Net method [36] is the first standardized out-of-the-box publicly available tool in biomedical segmentation. It is a self-adapting algorithm that selects the hyper-parameters, such as the batch size, patch size, and network topology, depending on the dataset given by the user with a set of heuristic criteria. nnU-Net offers a fully automated deep learning pipeline, including three different 3D U-Net architectures with a depth of 5. It selects the best network architecture through a 5-fold cross-validation procedure to split the data into training and validation sets. The same test set was withheld in this study, and the rest of the data were used for cross-validation. The estimated best performance of all nnU-Net models was the 3D full-resolution architecture [33]. Subsequently, the 3D full-resolution nnU-Net models were separately trained on the NACT and CECT datasets, with 32 feature channels and a batch size of 2 for comparison. More nnU-Net architecture details can be found in Figure 1.

#### 2.2.3. AI-Based Mouse Organ Segmentation (AIMOS)

Schoppe et al. [32] introduced AIMOS, a specialized deep-learning pipeline for segmenting mouse organs in micro-CT images. This system offers various 2D U-Net-like architectures with minimal user intervention. For this study, we employed the UNet-768 structure, featuring six encoder–decoder stages with 32 and 768 feature channels at the highest layer and the bottleneck, respectively. The network was trained on all slices with a batch size of 32. Previous studies have benchmarked nnU-Net performance against AIMOS [33] for thorax organs. We adapted the AIMOS pipeline, customizing the data split and preprocessing to train on the NACT dataset. This 2D-based AIMOS model was then compared with the other two models using the IHCT dataset.

### 2.3. Evaluation Metrics

The segmentation network performance was quantitatively evaluated using the Dice similarity coefficient (*DSC*) and the 95th percentile of the Hausdorff distance (*HD_95p_*) [52] between automated and manual reference contours. The analysis is performed for individual scans and then combined for each unique animal via averaging if multiple scans are present for the same animal.
(1)$$DSC=\frac{2\left|A⋂B\right|}{\left|A\right|+\left|B\right|}$$
(2)$$HD_{95p}=max\left\{d_{95}\left(A,\;B\right),\;d_{95}\left(B,A\right)\right\},\;\;\;d_{95}=\underset{a\in A}{x_{95}}\left\{\underset{b\in B}{\min}\;d\left(a,\;b\right)\right\}$$

*DSC* measures the volume overlap between the references and predicted masks. *A* and *B* represent the corresponding voxels of the ground truth and the prediction, respectively. $x_{95}$ denotes the 95th percentile. The *HD*_95*p*_ is a specific variant of the Hausdorff Distance, designed to be robust toward outliers yet relevant to radiation treatment planning, aiming at constraining most voxels to be within a certain dose level. The average *DSC* and *HD*_95*p*_ were used for analysis to provide a balanced representation of the data across mice with varying numbers of scans, ensuring that the results were not influenced disproportionately by those with more or fewer scans.

### 2.4. Implementation Details

In our project, all neural networks were trained using a single NVIDIA RTX A6000 with 48 GB of GPU memory. A five-fold cross-validation method was uniformly applied across all models. During each fold, two mice for NACT and one mouse for CECT were randomly selected for validation. The Adam optimizer with an initial learning rate of 0.001 was applied. The same dataset split configuration was used for all networks, with the test set and the external set withheld to evaluate and compare the predictions generated from all networks. The inference speed was evaluated on the same system. The loss curves for the three models were provided in Appendix A.

## 3. Results

Seven major mouse organs were segmented using the Swin UNETR, 3D nnU-Net, and AIMOS. The test sets withheld from the NACT and CECT datasets were inferred and analyzed separately, and the IHCT was used further to investigate each model’s robustness across institutional boundaries. Figure 2 and Figure 3 illustrate comparisons for median cases between manual contouring and automated contouring using the Swin UNETR model and the 3D nnU-Net model from the NACT and CECT test sets. Generally, both neural networks accurately segmented the target organs, except for the intestine and spleen. The spleen lacked contrast in the NACT dataset but was visible in the CECT dataset.

Additionally, the automatically delineated boundaries of predictions from both models were smoother compared to the actual ground truth, a characteristic most evident in the lungs. Figure 4 shows the model’s generalizability to the IHCT using completely different imaging equipment, protocols, and organ annotators. Swin UNETR was better at capturing lung features and providing more precise boundary predictions for the bladder, liver, and kidneys than the other two models.

Both the average DSC and HD_95p_ are reported on the individual animal level for the NACT and CECT test sets in Table 2 and Figure 5. For the NACT test set, Swin UNETR generally showed slightly higher DSC in most organs, except in the intestine, where nnU-Net performed marginally better in two mice. Consistently, both neural networks had difficulties with spleen segmentation in the NACT dataset, resulting in approximately 70% DSC and ~1 mm HD_95p_. However, for the CECT test set, contrast agents significantly enhanced spleen segmentation, achieving more than 90% DSC and ~0.6 mm HD_95p_. All models achieved HD_95p_ less than 1 mm except for the liver and intestine in both NACT and CECT test sets and the spleen in the NACT test set. For the IHCT dataset, metrics are reported for five mice in Table 3, and Figure 6 specifically compares the DSC performance for each individual mouse. Superior performance and generalization using Swin UNETR, compared to the other two models trained with the NACT dataset, were more pronounced in the completely unseen IHCT dataset. Swin UNETR consistently achieved superior DSC and HD_95p_ for all seven organs, providing more than 80% DSC in the bladder, lungs, and liver, while the other two models’ performance suffered an evident drop to ~70% DSC. For kidneys, the DSC also improved from ~65% for the 3D nnU-Net and AIMOS to 74.4% for Swin UNETR. The HD_95p_ was improved to 1.6 mm vs. 2.8 mm for AIMOS.

The yellow arrows in Figure 2, Figure 3 and Figure 4 denote key differences in segmentation performance across various models. Specifically, arrows in Figure 2, Figure 3 and Figure 4 indicate that all deep-learning models yield smoother boundary contours for the lungs and mis-segmentations in the intestine. In Figure 2 for the NACT dataset, arrows show over-segmentations at the spleen boundary due to a lack of contrast with adjacent tissues. In Figure 4 for the IHCT dataset, more false-positive islands can be observed in heart, liver, and kidney segmentation, as well as minor false-negative islands in the bladder.

Table 4 presents a comparison of the average training time per epoch and the inference speed for each model on the NACT dataset, with all values measured in seconds. Among the models evaluated, the 2D-based AIMOS demonstrated the most efficient performance, boasting the shortest training time at 17.8 s per epoch and the fastest inference speed at 4.4 s per scan. In contrast, Swin UNETR exhibited the longest training and inference durations, taking 172.5 s per epoch and 54.7 s per scan, respectively. This increased time can be attributed to the model’s use of a larger window size and overlap factor in this study.

## 4. Discussion

3D image-guided pre-clinical irradiation platforms afford more accurate and conformal dose delivery for targeted interventional response assessment. The conformal dose delivery also lends to more translatable radiation research [53]. Moreover, the versatility of these platforms for precise dose delivery is illustrated by van Hoof et al. [54], who proposed an advanced dose painting strategy to enhance target conformality. However, conformal dose distribution needs to be contextualized with 3D organ contours, which are not readily available in the existing pre-clinical research workflow. The importance of accurate organ and structure contours increases with the recently introduced small animal intensity-modulated radiotherapy (IMRT), which better mimics human radiotherapy [55,56,57,58,59,60]. IMRT planning solves an inverse optimization problem for specific 3D organ dosimetric goals, thus requiring accurate delineation of involved organs. Manual delineation of normal organs has been routinely performed for human patients, but such a task can be impractical for pre-clinical research. Automated segmentation of the mouse organs has been performed using conventional methods, including active contouring and deformable registration. DL has emerged as a more precise tool for organ segmentation in mouse imaging. Several existing studies have employed CNNs, e.g., 2D and 3D U-Net, to facilitate automated segmentation of mouse organs. Despite their improved performance over conventional methods, CNNs lack inherent self-attention mechanisms necessary for stable performance, as shown in “domain shift” problems [61]. Swin Transformers demonstrated superior robustness with the shifted window self-attention mechanism and a hierarchical architecture for natural image processing. In this study, we focused on investigating the robustness of Swin Transformers versus U-Net variants (nnU-Net and AIMOS) [32,36].

Swin UNETR consistently outperformed the 3D nnU-Net model except for two mice, where the latter performed marginally better on the intestine, which has an intrinsically large manual delineation uncertainty due to morphologically complex and heterogeneous imaging intensity with chyme [32,33]. This uncertainty was observed from inter- and intra-observer variations in the ground truth contours across the three test sets [32]. Additionally, intestine contouring can be highly subjective [32], lacking clear boundaries to surrounding tissues, especially in low-resolution and noisy CBCT images. Spleen segmentation posed a challenge for both neural networks in the NACT dataset, for the spleen often exhibits low contrast relative to adjacent tissues, such as the stomach, intestine, pancreas, and kidneys. Human contouring of the spleen is often less accurate in these cases, which directly impacts model training and leads to suboptimal inference. Notably, applying contrast agents in the CECT test set yielded significant improvements in spleen segmentation, achieving over 90% DSC, as illustrated in Figure 3, where the spleen appears highlighted. As shown in Appendix A, examples of native and contrast-enhanced CBCT scans demonstrate that Swin UNETR mitigates over-segmentation. However, the key factors affecting segmentation accuracy are the image quality and the use of contrast agents. Although contrast agents did influence the liver, there were no discernible improvements in the CECT dataset. In NACT images, the liver typically exhibits well-defined boundaries and relatively high contrast with adjacent tissues, which allows for accurate segmentation by neural networks. These intrinsic characteristics of the liver may be sufficient for precise segmentation, rendering the additional contrast agents used in CECT less impactful.

Domain shift is one of the crucial tests to measure DL model generalizability [61], a property essential for the pre-clinical micro-CT images demonstrating substantial inter-institution variation due to the lack of standards. Imaging equipment, scanning geometry, kVp, and mAs can substantially alter the CT image characteristics. Yet, existing models for mouse segmentation are often trained and tested on samples acquired within one institution with homogeneous scanning parameters. The untested model robustness can hinder the adoption of automated segmentation. We acquired the IHCT dataset using different micro-CT and image protocols to test the model robustness, including a lower kVp and a resultant lower SNR. A different expert annotated the images. Inter-observer variations led to an estimated average decline in DSC of 8% and HD_95p_ of 0.5 mm for Swin UNETR compared to a 15% and 1 mm decline for 3D nnU-Net. Both models were trained on the NACT dataset with the original annotator. The declines in DSC and HD_95p_ were calculated by comparing the models’ average performance on the IHCT dataset, annotated by a different individual, against their performance on the NACT dataset. This comparison provided an approximate measure of annotator bias, showing the performance drop when models trained on one annotator’s labels were evaluated against another. Swin UNETR demonstrated its robustness to domain shift and consistently outperformed 3D full-resolution nnU-Net and AIMOS. The superior performance of Swin UNETR can be firstly attributed to its ability to capture long-range dependencies through an inherent self-attention mechanism, allowing for accurate organ structure recovery. Second, the model effectively balances global context awareness with local feature extraction by integrating Swin Transformers with a U-shaped network, further facilitating overall and detailed contour representation understanding.

This study is not without limitations. First, the public dataset used in this study included 221 sequential images of only 28 mice. Although we reasonably assumed the anatomical similarity of these genetically homogeneous mice, training was not based on fully independent samples. To mitigate this limitation, we carefully split data based on individual mice to minimize interdependence among the training samples. In the current work, on the other hand, the concern of data dependence is largely answered by the independent test on IHCT. Second, all NACT, CECT, and IHCT dataset scans were conducted in the prone position. Mice set up in other postures, including supine and decubitus positions, likely require training a new model on corresponding CT images. Third, neural network performance is inherently task-specific and contingent upon the data used, as different DL methods yield varying results in disparate datasets. Fourth, auto-segmentation on low native contrast organs (spleen) and morphologically complex organs (intestine) can be unreliable. Possible solutions, including domain adaptation adversarial generative networks [62], could be explored to highlight spleen-related voxels to assist neural networks further. Fifth, the performance in the external set declines slightly, but it still effectively reduces manual contouring effort and variability. Fine-tuning the models with additional scans for a few epochs can efficiently improve performance. Lastly, the test results depend on the quality and consistency of manual annotation. Digitization errors, such as the rough boundaries of the manual lung contours, contributed to residual discrepancies that cannot be completely eliminated.

## 5. Conclusions

In this study, we assessed the performance of Swin UNETR in segmenting major mouse organs across different datasets. Swin UNETR consistently outperformed 2D and 3D U-Net. Most importantly, Swin UNETR demonstrated superior robustness via testing on an independent mouse CT dataset with substantially different image characteristics. The resilience to more noisy images is an important step toward a generalizable auto-segmentation method for pre-clinical radiation research.

Organ contouring is an indispensable step in preclinical radiation research. Accurate mouse organ segmentation facilitates key tasks in radiation biology, such as treatment planning and dose distribution analysis, to better understand biological outcomes. The proposed automated segmentation models offer significant improvements in reproducibility and efficiency for preclinical workflows. Based on our findings, these models are recommended for segmenting the bladder, lungs, heart, liver, and kidneys with minimal manual corrections. However, careful examination is necessary when applying them to segment the intestine and the spleen in native CBCT images.

## Figures and Tables

**Figure 1 bioengineering-11-01255-f001:**
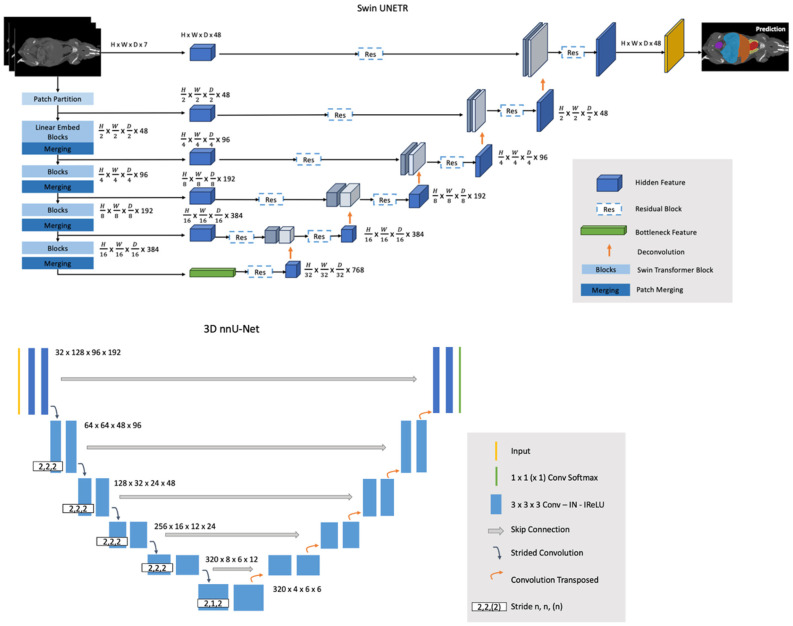
Swin UNETR architecture and 3D nnU-Net architecture used in this study.

**Figure 2 bioengineering-11-01255-f002:**
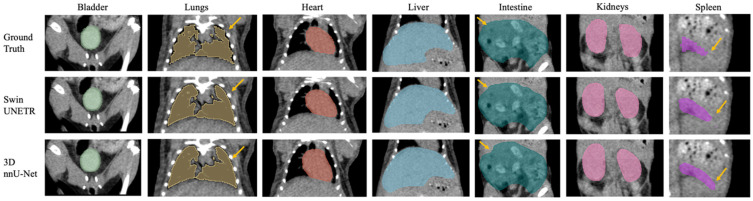
Example of the median-scored case in mouse multi-organ segmentation in coronal view from the NACT test set. Yellow arrows highlight key differences in segmentation outcomes between the two 3D models, Swin UNETR and nnU-Net.

**Figure 3 bioengineering-11-01255-f003:**
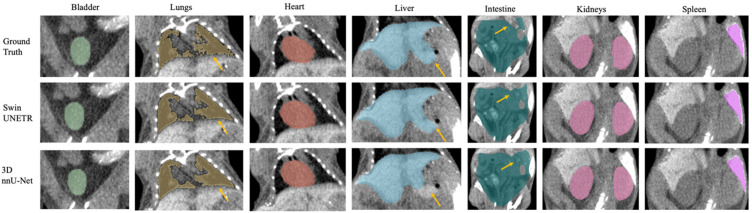
Example of the median-scored case in mouse multi-organ segmentation in coronal view from the CECT test set. Yellow arrows highlight key differences in segmentation outcomes between the two 3D models, Swin UNETR and nnU-Net.

**Figure 4 bioengineering-11-01255-f004:**
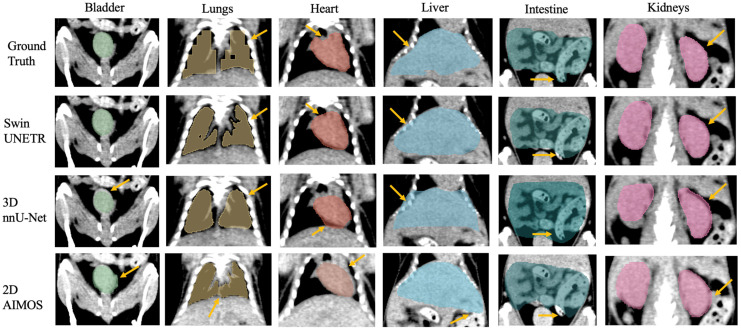
Example of the median-scored case in mouse multi-organ segmentation in coronal view from the IHCT test set. Yellow arrows highlight key differences in segmentation outcomes between the three models, 3D Swin UNETR, 3D nnU-Net, and 2D AIMOS.

**Figure 5 bioengineering-11-01255-f005:**
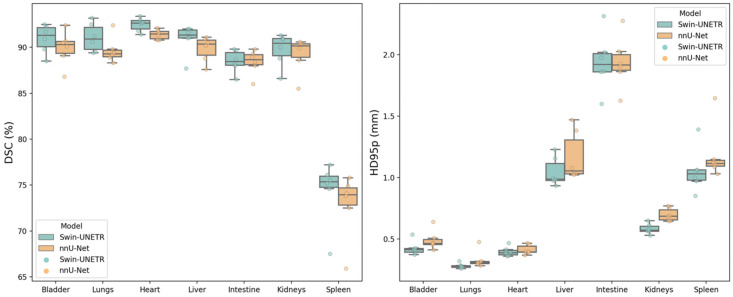
Box plots of DSC (%) and HD95p (mm) per organ for predictions by Swin UNETR (green) vs. nnU-Net (orange). Each box extends from the lower to the upper quartile values of the data, with a black line at the median; the whiskers extend to the outermost data point within 1.5 times the interquartile range.

**Figure 6 bioengineering-11-01255-f006:**
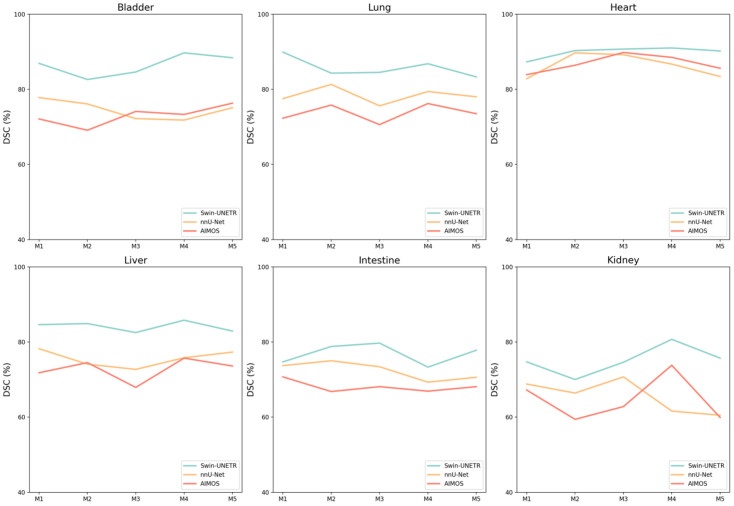
DSC (%) performance comparisons for each individual mouse in the IHCT test set.

**Table 1 bioengineering-11-01255-t001:** Details of three datasets used in this study.

Dataset	Source	Scanner	Number of Animals		Number of Images		Energy	Resolution
			Training/ Validation	Test	Training/ Validation	Test		
Native CT (NACT)	Public Dataset	Tomoscope Duo	14	6	98	42	65 kV	0.14 mm × 0.14 mm × 0.14 mm
Contrast-enhanced CT (CECT)	Public Dataset	InSyTe	6	2	66	15	75 kV	0.14 mm × 0.14 mm × 0.14 mm
In-house CT (IHCT)	City of Hope	SmART	/	5	/	5	40 kV	0.2 mm × 0.2 mm × 0.2 mm

**Table 2 bioengineering-11-01255-t002:** Quantitative evaluation results on average DSC (%) and HD95p (mm) for the NACT and CECT test sets. Swin UNETR model performance is compared with the 3D full-resolution nnU-Net model. The better results are bolded, with those not from Swin UNETR also underlined.

DSC (%) mean ± s.d.		Bladder-1		Lungs-2		Heart-3		Liver-4		Intestine-5		Kidneys-6		Spleen-7	
		Swin UNETR	nnU-Net	Swin UNETR	nnU-Net	Swin UNETR	nnU-Net	Swin UNETR	nnU-Net	Swin UNETR	nnU-Net	Swin UNETR	nnU-Net	Swin UNETR	nnU-Net
**NACT**	M01	**90.9 ± 1.8**	90.1 ± 2.3	**89.4 ± 1.5**	88.3 ± 1.3	**92.6 ± 1.3**	90.8 ± 1.2	**91.1 ± 1.2**	88.8 ± 1.3	**86.5 ± 2**	86 ± 1.8	**91 ± 1.1**	90.4 ± 1	**74.6 ± 7.7**	72.5 ± 8.7
	M02	**92.5 ± 2.3**	92.4 ± 2	**91.2 ± 0.9**	89.2 ± 0.9	**91.8 ± 1.2**	90.8 ± 1.3	**92 ± 1.3**	90.5 ± 1.1	**89.7 ± 1.9**	89.3 ± 1.8	**91.3 ± 1.4**	90.4 ± 1.3	**75.2 ± 3.6**	73.8 ± 6.5
	M03	**92.3 ± 0.6**	90.7 ± 1	**90.6 ± 1.2**	89.4 ± 1.5	**91.4 ± 0.9**	91.3 ± 1.1	**91 ± 0.5**	90.9 ± 1.1	88.7 ± 1.5	** 88.9 ± 1.2 **	**88.8 ± 2**	88.6 ± 3.1	**75.5 ± 3.9**	74.9 ± 4.2
	M04	**91.7 ± 1.6**	90.5 ± 1.5	**92.5 ± 0.7**	89.8 ± 0.9	**92.7 ± 0.6**	91.6 ± 0.5	**91.6 ± 1**	91.1 ± 1.2	89.8 ± 1.3	89.8 ± 1.5	**90.9 ± 1.1**	90.6 ± 1	**76.1 ± 4.4**	74.1 ± 6
	M05	**88.5 ± 0.6**	86.8 ± 1.3	**89.5 ± 0.7**	88.9 ± 0.9	**93.4 ± 0.3**	92.1 ± 0.4	**92 ± 1.3**	90.2 ± 1.4	88.2 ± 1.6	** 88.4 ± 1.7 **	**90 ± 1.4**	89.9 ± 1.7	**77.2 ± 2.8**	75.8 ± 4.5
	M06	**89.8 ± 2.6**	89.1 ± 2.3	**93.2 ± 1.6**	92.4 ± 1.5	**93 ± 0.6**	91.8 ± 0.8	**87.7 ± 0.7**	87.6 ± 1.8	88 ± 1.5	88 ± 2.2	**86.6 ± 2.7**	85.5 ± 3	**67.5 ± 7.2**	65.9 ± 8.7
CECT	M01	**91.9 ± 3**	90.1 ± 3.2	89.9 ± 4	89.9 ± 4.8	**93 ± 1.3**	92.3 ± 1.6	**92.7 ± 0.3**	92.4 ± 0.6	**89.5 ± 2.9**	88.9 ± 2.8	**92 ± 1.3**	91.2 ± 1.3	**92.4 ± 1.4**	90.5 ± 2.5
	M02	**91.4 ± 3.1**	89.1 ± 3	**86.6 ± 5.5**	85.8 ± 7.1	**92.5 ± 1.4**	90.8 ± 1.6	**88.5 ± 4.2**	85.5 ± 4.3	**84.9 ± 3.5**	84.6 ± 5.1	**88.8 ± 2.8**	87.7 ± 2.6	**92.3 ± 1.3**	91.4 ± 1
**HD_95p_(mm)** **mean ± s.d.**		Bladder-1		Lungs-2		Heart-3		Liver-4		Intestine-5		Kidneys-6		Spleen-7	
		Swin UNETR	nnU-Net	Swin UNETR	nnU-Net	Swin UNETR	nnU-Net	Swin UNETR	nnU-Net	Swin UNETR	nnU-Net	Swin UNETR	nnU-Net	Swin UNETR	nnU-Net
NACT	M01	**0.42 ± 0.16**	0.47 ± 0.16	**0.29 ± 0.11**	0.31 ± 0.16	**0.42 ± 0.2**	0.46 ± 0.24	**1.16 ± 0.47**	1.38 ± 0.33	1.86 ± 0.66	1.86 ± 0.53	**0.57 ± 0.2**	0.65 ± 0.11	**0.97 ± 0.32**	1.11 ± 0.44
	M02	**0.39 ± 0.17**	0.41 ± 0.17	0.28 ± 0.09	0.28 ± 0.08	0.47 ± 0.14	0.47 ± 0.16	**0.93 ± 0.37**	1.03 ± 0.39	**1.86 ± 0.46**	1.93 ± 0.68	**0.56 ± 0.18**	0.69 ± 0.12	**0.85 ± 0.19**	1.15 ± 0.29
	M03	**0.41 ± 0.14**	0.45 ± 0.13	**0.27 ± 0.07**	0.48 ± 0.11	**0.37 ± 0.15**	0.40 ± 0.16	**0.98 ± 0.28**	1.03 ± 0.62	2.31 ± 0.68	** 2.28 ± 0.7 **	**0.61 ± 0.29**	0.75 ± 0.24	**1.06 ± 0.19**	1.03 ± 0.17
	M04	**0.43 ± 0.12**	0.46 ± 0.13	**0.27 ± 0.12**	0.30 ± 0.14	0.39 ± 0.1	0.39 ± 0.17	**0.99 ± 0.3**	1.02 ± 0.38	**1.60 ± 0.4**	1.63 ± 0.58	**0.53 ± 0.14**	0.65 ± 0.15	**1.00 ± 0.23**	1.09 ± 0.34
	M05	0.64 ± 0.17	0.64 ± 0.19	**0.26 ± 0.1**	0.31 ± 0.15	**0.36 ± 0.07**	0.37 ± 0.08	**0.98 ± 0.29**	1.08 ± 0.3	1.98 ± 0.33	** 1.91 ± 0.45 **	**0.58 ± 0.18**	0.68 ± 0.17	**1.06 ± 0.21**	1.12 ± 0.32
	M06	**0.37 ± 0.14**	0.51 ± 0.16	0.32 ± 0.18	0.32 ± 0.2	0.39 ± 0.16	0.39 ± 0.2	**1.23 ± 0.53**	1.47 ± 0.66	**2.02 ± 0.51**	2.03 ± 0.62	**0.65 ± 0.15**	0.77 ± 0.25	**1.39 ± 0.3**	1.65 ± 0.45
CECT	M01	**0.38 ± 0.1**	0.44 ± 0.2	**0.25 ± 0.15**	0.29 ± 0.13	**0.52 ± 0.24**	0.58 ± 0.19	**1.14 ± 0.29**	1.26 ± 0.42	**1.78 ± 0.5**	1.94 ± 0.67	**0.62 ± 0.29**	0.65 ± 0.24	**0.72 ± 0.2**	0.76 ± 0.33
	M02	**0.46 ± 0.11**	0.50 ± 0.14	**0.27 ± 0.13**	0.35 ± 0.1	**0.59 ± 0.27**	0.62 ± 0.2	**1.06 ± 0.39**	1.10 ± 0.5	**2.04 ± 0.9**	2.10 ± 0.8	**0.59 ± 0.2**	0.83 ± 0.38	**0.64 ± 0.12**	0.73 ± 0.26

**Table 3 bioengineering-11-01255-t003:** Quantitative evaluation results on average DSC (%) and HD95p (mm) for the IHCT test set. Swin UNETR model performance is compared with the 3D full-resolution nnU-Net and the published AIMOS model. The better results are bolded.

	Bladder-1			Lungs-2			Heart-3			Liver-4			Intestine-5			Kidneys-6		
mean ± s.d.	Swin UNETR	nnU-Net	AIMOS	Swin UNETR	nnU-Net	AIMOS	Swin UNETR	nnU-Net	AIMOS	Swin UNETR	nnU-Net	AIMOS	Swin UNETR	nnU-Net	AIMOS	Swin UNETR	nnU-Net	AIMOS
DSC (%)	**86.4 ± 2.6**	74.6 ± 2.3	73.0 ± 2.4	**84.7 ± 2.4**	78.4 ± 1.9	73.7 ± 2.1	**89.9 ±** 1.3	86.3 ± 2.9	86.8 ± 2.1	**84.2 ±** 1.2	75.6 ± 2.0	72.7 ± 2.7	**76.8 ± 2.5**	72.4 ± 2.1	68.1 ± 1.4	**74.4 ± 3.4**	66.2 ± 4.8	64.6 ± 5.4
HD_95p_ (mm)	**0.6 ± 0.2**	0.8 ± 0.6	1.7 ± 1.1	**0.7 ± 0.2**	1.2 ± 0.1	1.0 ± 0.2	**0.6 ± 0.1**	1.1 ± 0.1	0.8 ± 0.1	**1.8 ± 0.2**	2.1 ± 0.6	2.2 ± 0.9	**2.6 ± 0.8**	2.8 ± 0.7	3.0 ± 1.7	**1.6 ± 0.5**	2.5 ± 0.9	2.8 ± 0.4

**Table 4 bioengineering-11-01255-t004:** Comparison of average training time per epoch and inference speed per scan, measured in seconds.

Model	Swin UNETR	3D nnU-Net	AIMOS
**Training time per epoch (s)**	172.5	57.0	17.8
**Inference (s)**	54.7	20.8	4.4

## Data Availability

The public data that support the findings of this study are openly available at the following URL: https://nature.com/articles/sdata2018294 (accessed on 21 November 2024). The pre-trained models and code for implementing this project are open sourced at: https://zenodo.org/records/13841139 (accessed on 21 November 2024).

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
