# Peer review of "Robust Automated Mouse Micro-CT Segmentation Using Swin UNEt TRansformers"

_bioengineering, 2024, doi:10.3390/bioengineering11121255_

Round 1

Reviewer 1 Report

Comments and Suggestions for Authors

After careful reading the article, the reviewer found the following suggestion which authors must rectify before moving to the next step.

(1)    The slices from the same volume of each animal may induce structural similarity bias in images. Authors must include some variation in the images by producing the augmented images.

(2)    However, authors mentioned about the data harmonization/homogenization do they observe any effect on organ segmentation with/without homogenization?

(3)    What is the model performance on combined NACT, CECT and IHCT database? Does it outperform the individual result?

(4)    In figure 4 authors must consider adding the markers of the datapoints to highlight the position of performance at each x-axis mark.

(5)    What is the need of multiorgan segmentation in animal write in objectives. Also, which organ do authors recommend to be segmented by their proposed model. Authors must write it in conclusion.

Reviewer 2 Report

Comments and Suggestions for Authors

The introduction mentions atlas-based segmentation and machine learning approaches but doesn't delve into specific shortcomings, such as performance metrics, generalizability issues.

The statement that fully convolutional neural networks (FCNNs) "are unstable for segmenting heterogeneous data outside the training cohort" is broad and lacks supporting evidence or references.

The test sets, particularly the IHCT dataset with only 5 mice, are very small to draw robust conclusions.

The section mentions difficulties in segmenting the spleen and intestine but does not delve into why these challenges occur. There's no qualitative or quantitative error analysis to understand the underlying causes of mis-segmentations.

There's no information on whether the models might be overfitting to the training data.

The results focus solely on performance metrics without considering how interpretable the models are.

Round 2

Reviewer 2 Report

Comments and Suggestions for Authors

The author answered all my questions. The manuscript has been sufficiently improved to warrant publication in Bioengineering.